# CXCL13 Positive Cells Localization Predict Response to Anti-PD-1/PD-L1 in Pulmonary Non-Small Cell Carcinoma

**DOI:** 10.3390/cancers16040708

**Published:** 2024-02-07

**Authors:** Fatemeh Vahidian, Fabien C. Lamaze, Cédrik Bouffard, François Coulombe, Andréanne Gagné, Florence Blais, Marion Tonneau, Michèle Orain, Bertrand Routy, Venkata S. K. Manem, Philippe Joubert

**Affiliations:** 1Centre de Recherche de l’Institut Universitaire de Cardiologie et de Pneumologie de Québec (IUCPQ), Quebec City, QC G1V 4G5, Canadafabien.lamaze@criucpq.ulaval.ca (F.C.L.); michele.orain@criucpq.ulaval.ca (M.O.); 2Faculty of Medicine, Laval University, Quebec City, QC G1V 4G5, Canadaflorence.blais.1@ulaval.ca (F.B.); 3Centre de Recherche du Centre Hospitalier de l’Université de Montréal (CRCHUM), Montréal, QC H2X 0A9, Canada; marion.tonneau.chum@ssss.gouv.qc.ca (M.T.);; 4Centre de Recherche du CHU de Québec—Université Laval, Quebec City, QC G1V 4G5, Canada; 5Department of Mathematics and Computer Science, Université du Québec à Trois-Rivières, Trois-Rivières, QC G8Z 4M3, Canada

**Keywords:** Non-small cell lung cancers (NSCLCs), CXCL13, tertiary lymphoid structure (TLS), tumor microenvironment (TME), immune checkpoint inhibitor treatments (ICIs)

## Abstract

**Simple Summary:**

Lung cancer is the most lethal cancer worldwide. Recently, immunotherapy has revolutionized the therapeutic landscape of pulmonary non-small cell carcinomas. While the utilization of immune checkpoint inhibitors targeting the PD-1/PD-L1 axis has improved survival, only a small percentage of cases will show a durable response. In this project, we aim to assess the potential of CXCL13+ cells to identify good responders to immunotherapy. Our results showed that the density and the localization of these cells are associated with the extent of the response to immune checkpoint inhibitors in advanced non-small cell lung cancers. This research provides new insights into the roles of the tumor microenvironment in modulating response to immunotherapy.

**Abstract:**

**Background:** Immune checkpoint inhibitors (ICIs) have revolutionized non-small cell lung cancers (NSCLCs) treatment, but only 20–30% of patients benefit from these treatments. Currently, PD-L1 expression in tumor cells is the only clinically approved predictor of ICI response in lung cancer, but concerns arise due to its low negative and positive predictive value. Recent studies suggest that CXCL13+ T cells in the tumor microenvironment (TME) may be a good predictor of response. We aimed to assess if CXCL13+ cell localization within the TME can predict ICI response in advanced NSCLC patients. **Methods:** This retrospective study included 65 advanced NSCLC patients treated with Nivolumab/Pembrolizumab at IUCPQ or CHUM and for whom a pretreatment surgical specimen was available. Good responders were defined as having a complete radiologic response at 1 year, and bad responders were defined as showing cancer progression at 1 year. IHC staining for CXCL13 was carried out on a representative slide from a resection specimen, and CXCL13+ cell density was evaluated in tumor (T), invasive margin (IM), non-tumor (NT), and tertiary lymphoid structure (TLS) compartments. Cox models were used to analyze progression-free survival (PFS) and overall survival (OS) probability, while the Mann–Whitney test was used to compare CXCL13+ cell density between responders and non-responders. **Results:** We showed that CXCL13+ cell density localization within the TME is associated with ICI efficacy. An increased density of CXCL13+ cells across all compartments was associated with a poorer prognostic (OS; HR = 1.22; 95%CI = 1.04–1.42; *p* = 0.01, PFS; HR = 1.16; *p* = 0.02), or a better prognostic when colocalized within TLSs (PFS; HR = 0.84, *p* = 0.03). **Conclusion:** Our results support the role of CXCL13+ cells in advanced NSCLC patients, with favorable prognosis when localized within TLSs and unfavorable prognosis when present elsewhere. The concomitant proximity of CXCL13+ and CD20+ cells within TLSs may favor antigen presentation to T cells, thus enhancing the effect of PD-1/PD-L1 axis inhibition. Further validation is warranted to confirm the potential relevance of this biomarker in a clinical setting.

## 1. Introduction

Non-small cell lung cancers (NSCLCs) are aggressive malignant neoplasms, and a large majority of patients with advanced disease progression have no possibility of curative surgical treatment [1,2]. Over the last decade, numerous clinical trials have demonstrated the survival benefit of immune checkpoint inhibitor treatments (ICIs), such as monoclonal antibodies that block immune checkpoints like programmed death-1 and programmed death-ligand 1 (PD-1/PD-L1), when used alone or in combination therapy [3,4]. While ICI treatment is associated with improved outcomes, a significant percentage (60–85%) of patients do not respond well [5]. 

The expression of PD-L1 in tumor cells is currently utilized as a biomarker to predict the response to anti-PD-L1/PD-1 therapies, although it is documented that some patients with high PD-L1 expression may experience worsened clinical outcomes and patients with low PD-L1 expression may show complete response [6]. Recent studies have revealed that some immune components of the tumor microenvironment (TME) could predict response to anti-PD-1/PD-L1 in several types of cancers [7,8]. The number of tertiary lymphoid structures (TLSs), which are ectopic lymphoid tissues that can develop in non-lymphoid organs, has emerged as a potential factor linked to a positive prognosis in diverse tumor types. Recent studies have unveiled the potential of TLSs in promoting immunotherapy response in melanomas, soft-tissue sarcomas, and renal cell carcinomas [9,10,11]. Moreover, researchers have highlighted the synergistic effects of TLS-associated B cells with T cells in exerting anti-tumor effects, and the co-presence of TLS and B cells in tumors has been found to enhance the immunotherapy effectiveness [9,10,11,12]. CXCL13s are chemokines controlling the recruitment of B cells to secondary lymphoid organs [13,14,15]. In cancer, it also plays a major role in the recruitment of B cells to TLSs [16,17,18,19,20], yet the potential effect of CXCL13 in the TME as a predictive prognostic biomarker is controversial. Several studies have demonstrated that high levels of CXCL13+ cells or CXCL13 expression in the TME could be used as a prognostic biomarker for ICI-treated bladder cancer and ovarian cancer patients and is associated with prolonged survival and objective response [21,22,23]. In addition, in another study, M. Sorin et al. revealed the effective role of CXCL13 as a soluble molecule within the TME of NSCLC patients and its potential in anti-PD-1 therapy [23]. In contrast, some studies have shown that CXCL13+ T cells were associated with inferior overall survival (OS) and progression-free survival (PFS) in treated renal cell carcinoma and gastric cancer patients [24,25]. In NSCLC, the impact of CXCL13+ cells in different compartments of the TME and its prognostic significance remains unclear.

In the current study, we investigated whether CXCL13+ cells could be used as a predictive biomarker for ICI response in NSCLC patients. In particular, we investigated whether CXCL13+ cell prediction performance was different in various compartments, including non-tumor (NT), invasive margin (IM), tumor (T), and TLS, even after accounting for other known predictive factors. We analyzed the density of CXCL13+ cells in those various compartments using whole sections from resection specimens. Additionally, by examining the CXCL13+ cells’ density in the different TME compartments in the presence of TLSs, we indirectly tested the hypothesis that CXCL13+ cells migrate toward the TLS compartment to recruit CD20+ cells within TLSs. 

## 2. Materials and Methods

### 2.1. Patient’s Clinical Data and Tissue Selection

This is a retrospective cohort study of patients with a diagnosis of NSCLCs, treated with Pembrolizumab/Nivolumab (2015 to 2021) in a first or second-line setting from the *Institut Universitaire de Cardiologie et de Pneumologie de Québec* (*IUCPQ*, n = 23) and the Centre Hospitalier de l’Université de Montréal (CHUM, n = 42). In order to be included in the current study, patients must have undergone a surgical resection prior to their immunotherapy treatment. A consent form was obtained for all participants by each institution’s biobank. Clinical and pathological data, including age, sex, smoking status, stage of cancer, and survival, were retrieved using the electronic medical files and pathology reports.

### 2.2. Immunohistochemistry

A thoracic pathologist (PJ) reviewed each tumor’s hematoxylin and eosin (H&E) histology slides, and one formalin-fixed paraffin-embedded (FFPE) block, representative of the lesion, was selected for each patient, ensuring it contained tumoral tissue. Sections of 4.0-μm-thick were cut from the selected blocks on a microtome and placed on charged slides. The following antibodies were used for IHC experiments: CXCL13, CD20, CD3, CD4, CD8, CD56, CD163, FoxP3, Lag3, TIM3, PD-1 (Appendix A). All slides underwent heat-induced epitope retrieval in a DAKO PT-Link using EnVision FLEX Target Retrieval Solution, high pH Tris/EDTA buffer (PH 9) (Dako, Agilent Technologies, Santa Clara, CA, USA), followed by an automatized IHC protocol on Dako Autostainer Link 48, using the Dako EnVision FLEX+ kit reagents (Dako, Agilent Technologies, Santa Clara, CA, USA). A previous fine-tuning assay with tonsil tissue enabled setting the CXCL13 antibody dilution to 1:500 and the incubation time to 20 min. Antibody clones and dilution protocols are available in Appendix A. After IHC, slides were counter-stained using EnVision FLEX hematoxylin reagent. 

PD-L1 score was performed prior to the initiation of the ICI treatment and extracted from clinical files. The assay used locally was antibody kit PD-L1 clone 22C3 (PharmDx, distributed by Agilent). Results were expressed in TPS (Tumor Proportion Score = % of tumor cells IHC positive) and separated into 3 categories: <1, ≥1 and <50, ≥50 or as a continuous variable. Characteristics of patients in relation to PDL1-TPS are presented in Appendix A.

### 2.3. Slide Digitalization and CXCL13 Scoring in Different Tumor Microenvironments

All H&E and IHC slides were digitalized at 20× magnification using a slide scanner (NanoZoomer 2.0-HT; Hamamatsu, Bridgewater, NJ, USA). CXCL13 IHC slides were visualized using the companion software NDP.view2 (Version 2.9.29) (Hamamatsu, Shizuoka, Japan). Prior to manual scoring CXCL13 and CD20, the compartments of the tumoral microenvironment were defined (Figure 1A). Briefly, they were annotated by a pathologist (PJ or CB) to determine Tumor (T), Invasive Margin (IM defined as 1 mm or less in size, including 500 µm in the tumor border line and 500 µm out of the tumor border line) and Non-Tumor (NT) areas. A pathologist (CB) identified the TLSs on CD20 IHC slides, with TLSs having a minimum of 50 aggregated CD20+ cells. Subsequently, the images of CXCL13 slides and CD20 slides were matched to locate identical viewpoints with precision, a process facilitated by the software NDP.view2. Scoring was completed manually by counting CXCL13+ cells within 1 mm^2^ areas of the T compartment (CXCL13_T) representing 10% of the respective investigated area, not overlapping with a TLS (>200 μm distance). We scored CXCL13+ cells localized in IM (CXCL13_IM), which represent almost half of the respective investigated area, and we scored CXCL13+ cells localized in NT (CXCL13_NT) representing 10% of the respective investigated area. The CXCL13+ cells were scored within the TLS (CXCL13_TLSin) and the TLS’s neighborhood (CXCL13_TLSne, 200 μm radius surrounding the TLS) for five randomly selected TLSs. Also, we computed the CXCL13+ cells count within and in the neighborhood of TLSs (CXCL13_TLS), as well as the TME area defined by the summation of CXCL13-T and CXCL13_TLS (CXCL13_TME) (Figure 1A). 

Since it has been shown that some CXCL13+ cells can either possess some of the following immune markers or be associated with immune cells expressing these immune markers (CD3, CD4, CD8, CD20, CD56, CD163, FoxP3, Lag3, TIM3, and PD-1) [24,25], those IHC markers were evaluated in the tumor compartment (including tumor and stroma) with the image analysis modules Tissue Recognition (Version 3.2.0) and Immunosurface (Version 3.0.0) from CaloPix software (CaloPix, Tribun Health, Paris, France). Both “tissue recognition” and “immune surface” applications were used to build an algorithm for the identification of tumor zones and segmentation of the IHC staining, as previously published [26]. Data were exported and compiled: the cellular positivity of a given IHC marker was expressed as the percentage of surface occupied by positive immune cells normalized by the surface of the analyzed area.

### 2.4. Survival Analysis

We investigated the prognostic significance of CXCL13+ cells in different compartments (Figure 1A), along with various other IHC markers (CD3, CD4, CD8, CD56, CD163, FoxP3, Lag3, TIM3, PD-1), and clinical and pathological features such as sex, age, Ecog performance status (Eastern cooperative oncology group), PD-L1-TPS, and histology groups. This analysis was carried out to evaluate if these markers and clinical features, along with CXCL13, could also predict the response to immunotherapy. A PFS (assessed through CT-scan results) of 1 year was defined as a cut-off based on previous studies to determine responders and non-responders [27,28,29,30,31], and PFS-months and OS-months were defined as the time between starting immunotherapy to the time of progression or last follow-up and death or last follow-up respectively. Univariate and multivariate COX models were employed to estimate hazard ratios (HRs) and 95% confidence intervals (CIs). Additionally, a comprehensive multivariate COX model was constructed, incorporating all the mentioned IHC markers and clinical features. A Spearman correlation coefficient between CXCL13 and other IHC markers was calculated and visualized with a heatmap. When two features were highly correlated, only one was kept for the following evaluations. To build the final multivariate COX model, we selected the variables with a two-step process: first, we selected significant features (*p* < 0.05) in the univariate COX models (26 variables), and second, we used a backward selection procedure within a multivariate COX model. We reported the HR for each COX model as well as the global HR for multivariate COX regressions.

### 2.5. Statistical Analysis

A Spearman correlation was carried out to test the association between CXCL13+ cell density in different compartments and OS/PFS, using the SciPy library and SciPy Stats from Python packages (Version 3.10). The Mann–Whitney test was used to examine the differences in the density of CXCL13+ cells among various compartments in patients categorized by the number of TLSs. Patients in terms of the number of TLSs were equally classified into four groups: those without TLSs (n = 0), low TLSs (n = 1–2), intermediate TLSs (n = 3–8), and high TLSs (n ≥ 9). The analysis involved Python libraries, including SciPy (Version 1.7.1), Matplotlib (Version 3.4.0), Pandas (Version 1.3.3), and Seaborn (Version 0.11.2), in addition to the SciPy Stats module (Version 1.7.0) from the Python package (Version 3.10). All analyses were conducted with a statistically significant threshold of 5% (*p* < 0.05) using a two-tailed approach. Both illustrations and statistical analyses were performed using the Python Jupiter program for Windows (Version 3.10).

## 3. Results

### 3.1. Clinical Features of Treated NSCLC Patients

The clinicopathological features of the 65 treated patients included in our cohort, treated either with Pembrolizumab (n = 27), Nivolumab (n = 31), or combinations of Nivolumab/Pembrolizumab and/or chemotherapy (n = 7) are described in Table 1. The number of males (n = 31) and females (n = 34) was similar, and the 1-year PFS rate was 43%; no significant differences based on gender were observed. The median age at surgery was 68 years. Adenocarcinoma was the most common histologic subtype (n = 54), followed by squamous cell carcinoma (n = 11) (Table 1). The other clinical and immunological features of the patients by PDL1-TPS subgroups are presented in Appendix A, respectively. 

### 3.2. Association between CXCL13+ Cell Density and Response to ICI

We first tested the linear association between CXCL13+ cell density across the TME compartments and OS or PFS in months in the whole cohort. We found a negative correlation (r = −0.25, *p* = 0.046) between OS and CXCL13+ cell density in the tumor compartment (T) (Figure 1B). We observed a positive correlation between the CXCL13_TLS/CXCL13_T ratio and OS (r = 0.41, *p* = 7 × 10^−4^) or PFS (r = 0.34, *p* = 0.006) (Figure 1C). On the other hand, we found a significantly negative correlation between CXCL13+ (T+NT) cells and OS (r = −0.3, *p* = 0.03) (Figure 1B), and PFS (r = −0.28, *p* = 0.04) (Figure 1C). All linear regression models are described in Appendix A.

Secondly, we investigated if the responders (PFS ≥ 1 year) and non-responders (PFS < 1 year) displayed differences in the density of CXCL13+ cells in each TME compartment. We observed that responders had a significantly lower CXCL13+ cell density in the tumor (*p* = 0.016), even considering the surrounding area of the tumor (CXCL13+ (T+IM), *p* = 0.019; or CXCL13+ (T+IM+NT), *p* = 0.003; or CXCL13+ cells (T+NT), *p* = 1.5 × 10^−3^) (Figure 2A–D). However, we found a greater density of CXCL13+ cells in TLSs compared to the tumor (CXCL13_TLS/CXCL13_T ratio, *p* = 0.031 (Figure 2E)) in responders vs. non-responders, which is believed to be influenced by interaction between CXCL13+ and CD20+ cells within the TLS compartment [9,22]. 

### 3.3. CXCL13 Is a Factor Predictive of Survival and Progression-Free Survival

Given that the density of CXCL13+ cells was associated with PFS and OS in certain compartments, we further investigated if CXCL13+ cells could predict OS and PFS in different TME compartments using univariate and multivariate COX models (Table 2, Table 3, Table 4 and Table 5). Our analysis revealed that CXCL13+ cells in NT and CXCL13+ (T+IM+NT) cells compartments slightly increased the risk of death (HR = 1.13, 95% CI [1.02–1.26]; HR = 1.22, 95% CI [1.04–1.42]) and progression (HR = 1.31, 95% CI [1.04–1.64]; HR = 1.16, 95% CI [1.02–1.32]) respectively in a univariate COX model (Table 2), but not in a multivariate model (Table 3). Interestingly, the CXCL13_TLS/CXCL13_T ratio predicted a decreased risk of progression (HR = 0.84, 95% CI [0.73–0.98]) in a univariate COX model (Table 4) but not in a multivariate model (Table 5). We reported the HR for each COX model as well as the global HR for multivariate COX regressions. We also found that while CXCL13+ (T+NT) cells could be used as an independent prognostic predictor for OS (HR = 1.23, 95% CI [1.07–1.42]) or PFS (HR = 1.15, 95% CI [1.02–1.30]), it was highly correlated with CXCL13+ (T+IM+NT) cells and thus was removed from multivariate COX regressions. The HR for all compartments is reported in Appendix A. Overall, these results suggested that CXCL13+ cells in collaboration with CD20+ cells in TLSs decreased the risk of progression, while in other compartments, the CXCL13+ cells in the absence of CD20+ cells increased the risk of death and progression. It is important to note that among other IHC markers tested (such as CD3, CD4, CD8, FoxP3, Tim3, Lag3, PD-1) (Appendix A) as well as the clinical features investigated (Appendix A), none were able to predict OS and PFS. In addition, to assess the potential role of all clinical and immunological variables (IHC markers) in predicting OS and PFS, we performed a multivariate COX model, and we obtained a non-significant global HR for OS (HR = 0.71, *p* = 0.50) and for PFS (HR = 0.42, *p* = 0.56), respectively (Appendix A). Multivariate COX analysis revealed that neither clinical nor immunological factors could independently predict OS or PFS.

### 3.4. CXCL13+ Cells in the Tumor Have a Link with Chemotaxis toward TLS

Our investigation focused on two aspects: first, understanding the notable increase in CXCL13+ cell density when transitioning from NT to T regions, and second, examining the transition in CXCL13+ cell density from T to TLSs in the presence of TLSs. We explored the idea that CXCL13+ cells move into the TLS to interact with CD20+ cells, potentially triggering the recruitment of additional immune cells. In order to test this prediction, we equally categorized patients into four groups based on TLS number (none, low, intermediate, and high). The characteristics of these groups are shown in Appendix A. Then, we compared the density of CXCL13+ cells in each compartment and group. We found that for patients without TLSs, the CXCL13+ cell density was significantly higher in IM compared to patients with low TLSs (*p* = 0.02), intermediate TLSs (*p* = 0.005), and high TLSs (*p* = 0.002) (Figure 3). Based on these results, we concluded that when at least two TLSs are present (the presence of only one TLS was not significant and did not seem logical for this comparison, Appendix A), CXCL13+ cells had a chemotactic link and migrated toward T and finally to the TLS compartment. Interestingly, we did not find any significant differences in the density of CXCL13+ cells in IM between patients with low, intermediate, and high TLSs, suggesting that having a minimum of two TLSs is adequate for CXCL13+ cells to have a migratory from NT toward to IM, T, and finally to TLS compartments (Figure 3).

## 4. Discussion

Until the recent emergence of immunotherapies, the treatment of advanced-stage NSCLC patients remained relatively unchanged. Although ICIs as immunotherapy have shown encouraging response rates in NSCLC patients, there is still a need to better select patients that are most likely to benefit from these expensive treatments. This can be achieved by identifying and applying context-dependent biomarkers with high predictive power to ICI treatments. By utilizing such biomarkers, we can maximize the therapeutic outcomes and tailor treatments more precisely to individual NSCLC patients [32]. While several predictive immune biomarkers for ICI have been proposed, including PD-L1 [5] and INF-ɣ [33], CXCL13 appears to exhibit superior performance, although there are conflicting reports regarding its impact on survival [21]. In this study, we hypothesized that the density of CXCL13+ cells predicted the response to ICI in NSCLC patients and that the prognostic potential of CXCL13+ cells varied based on their localization in the different TME compartments. Our results indicate that CXCL13+ cells are prognostic in the advanced-stage NSCLC patients treated with Pembrolizumab/Nivolumab. These findings are consistent with previous reports identifying CXCL13 as a prognostic marker for ICI response in several cancer types [21,24,34]. In this study, we investigated the long-term response by defining our responders versus non-responders using a one-year period, as previously published, which was associated with an Objective Response Rate (ORR) of 43.08% at 6 months [27,28,29,30,31]. The response rate is similar to previous studies using immunotherapy in NSCLC in a first- or second-line setting [35,36,37,38]. 

In addition, we have reported, for the first time, both a favorable and unfavorable association between improved clinical outcomes in NSCLC patients undergoing ICI treatment and the density of CXCL13+ cells, depending on their location within the TME and its adjacent area. This study conducted a thorough analysis of cell densities and investigated the localization of CXCL13+ cells in various compartments of NSCLC. This aspect has not been explored in previous studies. We found an unfavorable association between improved clinical outcomes in ICI-treated NSCLC and an elevated density of CXCL13+ cells in non-TLS regions, including the tumor, non-tumor, and invasive margin, as well as solely the NT (T+NT+IM, NT). Accordingly, a rise in the CXCL13+ cell density within the mentioned compartments increases the risk of death and tumor progression. Our findings are consistent with the prior study of Panse et al., which reported a strong relationship between CXCL13 expression and tumor progression in breast cancer by the ERK signaling pathway [39]. Previous studies did not assess the potential predictive role of CXCL13+ cells in IM, but Bindea et al. revealed that the absence of CXCL13 in individuals with colorectal tumors led to a decrease in the concentration of B cells in the invasive border of the tumors, resulting in a substantial increase in the likelihood of disease recurrence [40].

In our study, we have discovered a significant association between improved clinical outcomes in NSCLC patients treated with immunotherapy and an increased density of CXCL13 within TLSs compared to the tumor itself. Similarly, another study revealed a prognostic and predictive role of CXCL13-mediated TLSs in bladder cancer patients treated with ICI and favorable outcomes [21], but they did not evaluate CXCL13+ cell density potential role in different compartments. However, our findings are supported by substantial evidence that demonstrates the connection between CXCL13 and the formation of TLSs [16,41,42], as well as the recruitment of B cells, TFH-cells, and dendritic cells to the TLSs. These cells play a crucial role in presenting antigens to effector T cells, thereby influencing the loco-regional anti-tumor immunity of tumors [43,44]. Several studies have demonstrated that the presence of B cells in TLSs enhances the effectiveness of immunotherapy [45]. This fact was experimentally confirmed, wherein the inhibition of B cells by anti-CD20 resulted in a reduction of inflammation and a decrease in the number of CD8+ T cells [46]. Notably, TLSs with a high abundance of antigen-presenting cells exhibit better survival rates [44], and these findings underscore the vital role of TLSs in priming immune cell sites with antigen-presenting cells to help coordinate an effective immune response [34].

Based on our results, we believe CXCL13+ cells have a propensity to migrate from NT to IM and toward TLSs in patients who have at least two TLSs in their tumor area and have different functions in those different locations.

According to previous studies, CXCL13+ cells may have two different and opposite functions. When bound to their receptors, they have the ability to directly impact tumor growth by promoting the proliferation of cancer cells and inhibiting apoptosis [47,48,49,50]. In addition, CXCL13+ cells can indirectly promote the growth of tumor cells by facilitating their evasion from immune responses mediated by effector T cells and by attracting immune cells that have an immunosuppressive function [51,52,53]. This effect is mediated through the secretion of the immunoregulatory cytokine IL-10 by tumor cells or the recruitment of immunosuppressive cells such as myeloid-derived suppressor cells (MDSCs) and T-regulatory (Treg) cells within the TME, leading to a pro-tumoral activity. However, it is important to note that the CXCL13 axis also has the potential to elicit anti-tumoral responses by promoting the formation of TLSs [24], as pointed out before. 

Our study has a few limitations that need to be acknowledged. It is a retrospective study with a cohort of patients who underwent surgical resection at different intervals of time, which may impact the profile of the immune cells given the dynamic variation of the TME over time. In addition, the validation of our findings in an independent set of patients would strengthen our findings. This limitation also emphasizes the small size of our cohort, which undoubtedly reduces the statistical power to detect associations. Finally, double IHC staining was not used, so the exact subtype of CXCL13-producing cells could not be confirmed with specific marker.

## 5. Conclusions

Our findings provide insights into the distribution and specific localization of CXCL13+ cells within different areas of NSCLC patients’ tumor sites, including the IM, TLS, and NT regions. Notably, our study demonstrates that IHC analysis of CXCL13+ cells can predict favorable and unfavorable prognostic in NSCLC patients undergoing Pembrolizumab/Nivolumab treatment. Furthermore, the precise location of CXCL13+ cells rather than their number determines their function and role as either favorable or unfavorable prognostic indicators of treatment response. The diverse localization of CXCL13+ cells occurs through a process of movement, starting from the NT and progressing to the IM, T, and eventually to the TLSs. Additionally, utilizing these markers, we have developed risk score models that hold promise in determining the prognosis of NSCLC patients. Further studies are required to further evaluate and characterize CXCL13+ cells within specific compartments in a larger and more diverse cohort of patients to better understand the biological mechanisms driving the efficacy of immunotherapy. 

## Figures and Tables

**Figure 1 cancers-16-00708-f001:**
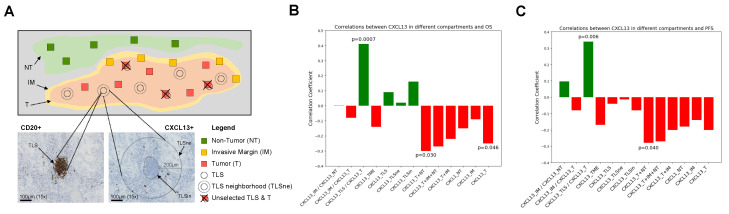
Schematic and histologic evaluation of TLS as well as CXCL13+ cells in different compartments. (**A**) IHC slide for assessing CXCL13+ cells in different compartments, including T, IM, TLS, and NT. (**B**) Spearman correlation coefficient in different compartments between CXCL13+ cells and OS, and (**C**) between CXCL13+ cells and PFS. T, tumor; IM, invasive margin; TLS, tertiary lymphoid structures; TLSin, inside TLS; TLSne, neighborhood of TLS; NT, non-tumor. TME, tumor microenvironment. A green color indicates a positive correlation whereas a red color indicates a negative correlation.

**Figure 2 cancers-16-00708-f002:**
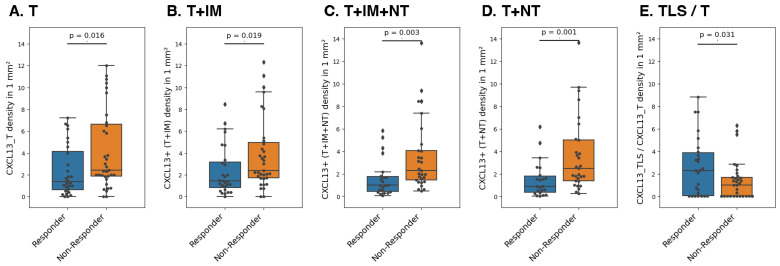
Comparison of CXCL13+ cell density between responders (PFS > 1 year) and non-responders (PFS < 1 year). Patients were treated with either Pembrolizumab or Nivolumab. In responders, a consistent lower density of CXCL13+ cells in non-TLS compartments are observed, such as T (**A**), T+IM (**B**), T+IM+NT (**C**), and T+NT (**D**). A higher density of CXCL13+ cells is observed in the TLS compared to the T compartments in responders (**E**). T, tumor; IM, invasive margin; NT, non-tumor.

**Figure 3 cancers-16-00708-f003:**
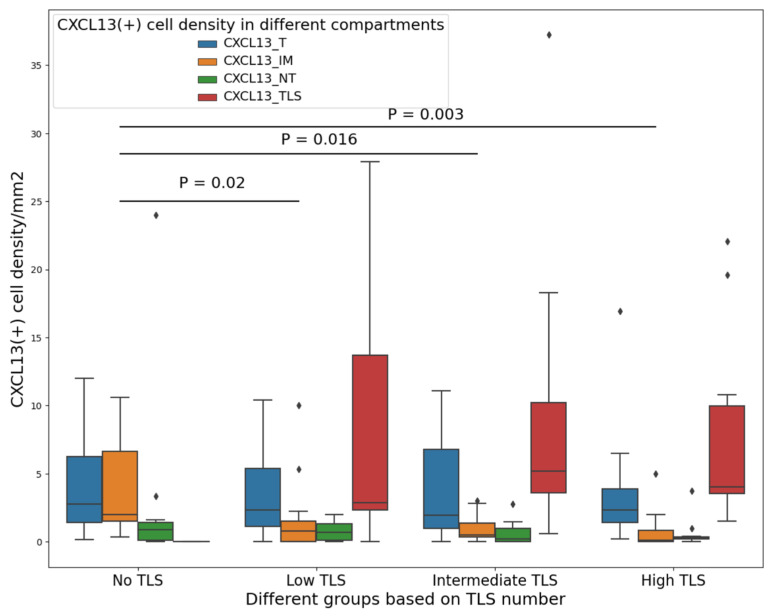
CXCL13+ cell density underly a migratory pattern toward the TLS compartment. This result shows CXCL13+ cells in patients with no TLS; the density of CXCL13+ cells in IM is significantly higher than in other groups (Low, Intermediate, and High groups). In cases with no TLS, CXCL13+ cells transition from NT to T and accumulate in IM. But in patients with at least 2 TLSs, CXCL13+ cells accumulate in TLSs after this transition. Low TLSs (1–2), intermediate (3–8), High (>=9). T, tumor; IM, invasive margin; NT, non-tumor; TLS, tertiary lymphoid structures.

**Table 1 cancers-16-00708-t001:** Clinicopathological characteristics of the selected 65 NSCLC patients.

Variables	Number of Patients (%)
**Sex** Male Female	31 (48%) 34 (52%)
**Smoking habit** Former Current Never	49 (75%) 14 (22%) 2 (3%)
**Histology groups** Adenocarcinoma Squamous	54 (83%) 11 (17%)
**Ecog-status** 0 1 2 3	18 (28%) 41 (63%) 4 (6%) 2 (3%)
**Type of immunotherapy** Nivolumab Pembrolizumab Pembrolizumab + chemotherapy Nivolumab + Pembrolizumab	31 (48%) 27 (42%) 6 (9%) 1 (1%)
**Stage at the start of immunotherapy** II III IV	2 (3%) 6 (9%) 57 (88%)
**PFS** >1 year <1 year	28 (43%) 37 (57%)

Ecog, Eastern cooperative oncology group; PFS, progression-free survival.

**Table 2 cancers-16-00708-t002:** Univariate COX model for CXCL13 positive cells in different compartments and OS.

Variables	Number of Patients	HR	CI 95% Lower	CI 95% Upper	*p*
CXCL13-NT	52	1.13	1.02	1.26	0.02
CXCL13-T+IM+NT	52	1.22	1.04	1.42	0.01
CXCL13-T+NT	52	1.23	1.07	1.42	<0.005

Abbreviations: T, tumor; IM, invasive margin; NT, non-tumor.

**Table 3 cancers-16-00708-t003:** Multivariate COX model for CXCL13 positive cells in different compartments and OS.

	**Overall Model HR**		**Overall Model *p***		
	1.23		0.08		
**Variables**	**Number of Patients**	**HR**	**CI 95%** **Lower**	**CI 95%** **Upper**	** *p* **
CXCL13-NT	52	1.05	0.92	1.20	0.49
CXCL13-T+IM+NT	52	1.18	0.98	1.41	0.09

Abbreviations: T, tumor; IM, invasive margin; NT, non-tumor.

**Table 4 cancers-16-00708-t004:** Univariate COX model for CXCL13 positive cells in different compartments and PFS.

Variables	Number of Patients	HR	CI 95% Lower	CI 95% Upper	*p*
CXCL13-NT	52	1.31	1.04	1.64	0.02
CXCL13-T+IM+NT	52	1.16	1.02	1.32	0.02
CXCL13-T+NT	52	1.15	1.02	1.30	0.02
CXCL13-TLS/CXCL13-T	63	0.84	0.73	0.98	0.03

Abbreviations: T, tumor; IM, invasive margin; TLS, tertiary lymphoid structures; NT, non-tumor.

**Table 5 cancers-16-00708-t005:** Multivariate COX model for CXCL13 positive cells in different compartments and PFS.

	**Overall Model HR**		**Overall Model *p***		
	1.16		0.56		
**Variables**	**Number of Patients**	**HR**	**CI 95%** **Lower**	**CI 95%** **Upper**	** *p* **
CXCL13-NT	50	1.25	0.99	1.57	0.06
CXCL13-T+IM+NT	50	1.05	0.89	1.23	0.56
CXCL13-TLS/CXCL13-T	50	0.89	0.75	1.05	0.17

Abbreviations: T, tumor; IM, invasive margin; TLS, tertiary lymphoid structures; NT, non-tumor.

## Data Availability

Data presented here are available from the corresponding author Philippe Joubert.

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
