# Peer review of "CXCL13 Positive Cells Localization Predict Response to Anti-PD-1/PD-L1 in Pulmonary Non-Small Cell Carcinoma"

_cancers, 2024, doi:10.3390/cancers16040708_

Round 1

Reviewer 1 Report

Comments and Suggestions for Authors

1. it could be better if use survive curve instead of low OS group vs high OS group

Author Response

Thank you very much for taking the time to review this manuscript. I have attached a Word file in response to your comment.

Reviewer 2 Report

Comments and Suggestions for Authors

There are few studies on CXCL13 in NSCLC, and the article is more innovative, but there are many problems:

1.     This article mainly explores the potential role of CXCL13+ cells to predict the efficacy of immunotherapy, and the main function of CXCL13 is to induce B lymphocyte homing. However, the authors neither introduce the function of CXCL13 in detail, nor introduce the relationship between B cells and the efficacy of NSCLC immunotherapy in the article, lacking research significance.

2.     The conclusion of CXCL13+ cells in predicting the efficacy of ICI was not verified in the validation set, which made the conclusions less reliable.

3.     The purpose of IHC to detect the expression of CD3, CD4, CD8, CD56, CD163, FoxP3, Lag3, TIM3 and PD-1 and its association with CXCL13 were unclear.

4.     In this paper, it was too simple that only PFS>1 year and < 1 year were used as the indicators for efficacy evaluation, and it was suggested to add ORR and other efficacy evaluation indicators to make the conclusion more convincing.

5.     The author's basis for classifying OS as cut off value of 1 year in Table 1 is not clarified and lacks scientific validity.

Comments on the Quality of English Language

The article needs to be moderately edited and improved in English.

Author Response

Thank you very much for taking the time to review this manuscript. 

I have attached a Word file in response to your nice comments. 

Also, the conclusion and introduction were improved. The English language was assessed.

Reviewer 3 Report

Comments and Suggestions for Authors

Comments for the authors:

In the manuscript “CXCL13 positive cells localization predict response to anti-PD- 1/PD-L1 in pulmonary non-small cell carcinoma”, the authors evaluated 65 advanced NSCLC patients treated with Nivolumab/Pembrolizumab and reported that different locations of CXCL13-positive cells have different prognostic implications. Authors showed the increase density of CXCL13-positive cells across all compartments were associated with a poorer prognosis, however, the increased density of CXCL13-positive cells within TLSs compared to the tumor itself were associated with a better prognosis. The data shown in this study suggest that the concomitant proximity of CXCL13+ and CD20+ cells within TLS may enhance antigen presentation to T cells, thus enhancing the effect of PD-1/PD-L1 axis inhibition. This study indicates that the prognostic potential of CXCL13+ cells vary based on its localization in the different TME compartments. This is an interesting study; however, the following issues can be solved. I made comments which I hope would benefit to this study.

Major comments:

1) The authors manually scored CXCL13-positive cells within limited areas, but this method lacks objectivity in verifying the significance of the differences in localization of CXCL13-positive cells. Recent technology has enabled an automated objective analysis over whole slide (Int J Mol Sci. 2022 Nov 8;23(22):13723. doi: 10.3390/ijms232213723.), which seems to be an appropriate method to support the authors' statements. This is a limitation of this study.

2) The authors should describe details of how the TLSs categorized into four group.

3) The patients’ characteristics according to TLS four groups should be shown. 

4) The characteristics of patients regarding to PD-L1 TPS should be shown. This study lacks the information of PD-L1 TPS. 

5) CXCL13 is expressed in the exhausted CD4+ and CD8+ T cells (Int J Mol Sci. 2022 Nov 8;23(22):13723. doi: 10.3390/ijms232213723.). CXCL13 expression is also shown to associate with immune-related adverse events (Proc Natl Acad Sci U S A. 2022 Jul 19;119(29):e2205378119.). The authors should discuss about what kind of cells the authors detect in this study express CXCL13. 

6) the authors should describe the limitations of this study.

7) Patients consent or ethical approval are not described. Is this study approved by IRB?

Author Response

I have applied your comment. Please find my responses in the attached file.

Round 2

Reviewer 2 Report

Comments and Suggestions for Authors

1.     The conclusion of CXCL13+ cells in predicting the efficacy of ICI was not verified in the validation set, which made the conclusions less reliable.

2.     Authors still not explained the reason of using PFS>1 year and < 1 year as the indicators for efficacy evaluation, and it was suggested to add ORR and other efficacy evaluation indicators to make the conclusion more convincing.

3.     Authors still not illustrated the reason of classifying OS as cut off value of 1 year in Table 1, which lacked scientific validity.

Comments on the Quality of English Language

The article needs to be moderately edited and improved in English.

Author Response

THank you for the great comments. I/we have uploaded our responses to your nice comments.

Reviewer 3 Report

Comments and Suggestions for Authors

The authors have made significant and appropriate changes to the manuscript and figures as per reviewers' suggestions. 

Author Response

Thank you for your nice comments, as you mentioned I/we addressed your nice comments the previous time.